# On Connective Tissue Mast Cells as Protectors of Life, Reproduction, and Progeny

**DOI:** 10.3390/ijms25084499

**Published:** 2024-04-19

**Authors:** Klas Norrby

**Affiliations:** Department of Pathology, Institute of Medical Biology, Sahlgren Academy, University of Gothenburg, 7 Ostindiefararen, SE-417 65 Gothenburg, Sweden; klas.norrby@pathology.gu.se; Tel.: +46-709146329

**Keywords:** angiogenesis, bFGF, cell proliferation, health, heparin, histamine, IL-1-alpha, IL-8, inflammation, nitric oxide, ovulation, pregnancy, proteases, sentinel cell, serotonin, survival, tissue remodeling, TNF-alpha, VEGF-A, wound healing

## Abstract

The connective tissue mast cell (MC), a sentinel tissue-residing secretory immune cell, has been preserved in all vertebrate classes since approximately 500 million years. No physiological role of the MC has yet been established. Considering the power of natural selection of cells during evolution, it is likely that the MCs exert essential yet unidentified life-promoting actions. All vertebrates feature a circulatory system, and the MCs interact readily with the vasculature. It is notable that embryonic MC progenitors are generated from endothelial cells. The MC hosts many surface receptors, enabling its activation via a vast variety of potentially harmful exogenous and endogenous molecules and via reproductive hormones in the female sex organs. Activated MCs release a unique composition of preformed and newly synthesized bioactive molecules, like heparin, histamine, serotonin, proteolytic enzymes, cytokines, chemokines, and growth factors. MCs play important roles in immune responses, tissue remodeling, cell proliferation, angiogenesis, inflammation, wound healing, tissue homeostasis, health, and reproduction. As recently suggested, MCs enable perpetuation of the vertebrates because of key effects—spanning generations—in ovulation and pregnancy, as in life-preserving activities in inflammation and wound healing from birth till reproductive age, thus creating a permanent life-sustaining loop. Here, we present recent advances that further indicate that the MC is a specific life-supporting and progeny-safeguarding cell.

## 1. Introduction

The connective tissue mast cell (MC) is a phylogenetically ancient secretory immune cell found in all vertebrate classes since approximately 500 million years, prior to the advent of adaptive immunity [1,2,3]. Notably, MC predecessors and current MCs contain heparin, histamine, and proteases, and, like human and rodent MCs, the early predecessors are activated by the MC secretagogue Compound 48/80 [1,2,3,4,5], suggesting significant consistent abilities of the MCs during a myriad of years. Distinct features of the MC are its unique composition of mediators and that it strongly affects the vasculature to accelerate selective cell recruitment when preparing for relevant required responses [6]. MCs are foremost known for their key role in IgE-mediated allergic reactions, including asthma, and anaphylaxis, the maximal variant of an allergic reaction involving several organ systems with risk of cardiovascular collapse. MCs are also activated by diverse immune and non-immune cells through secreted inflammatory mediators [7]. Nevertheless, accumulating evidence in the literature during the last decades speaks to a fundamental protective role for the MC in health and reproduction [4,7,8,9,10,11], as in the recently proposed perpetual survival of vertebrates [12]. Aspects of our present understanding of the complex life-promoting and life-sustaining influences exerted by the multifaceted MCs are, here, reviewed in brief.

## 2. The Mast Cell

The mast cell is a tissue-residing secretory long-lived immune cell featuring up to many hundreds of cytoplasmic-mediator-containing granules. It harbors numerous surface receptors, which enable its activation via a multitude of stimuli. The key growth factor for MCs is SCF (stem cell factor), the ligand for *Kit*, which is required for MC progenitor chemotaxis, differentiation, proliferation, and survival [13]. It is reported that human MCs can both produce and release SCF [14,15]. Upon activation, the MC releases a uniquely wide profile of preformed and de novo synthesized bioactive molecules. For instance, human MCs can release at least 390 different mediators and mouse MCs can release another 35 messenger substances [16]. These include heparin, histamine, serotonin, cytokines, chemokines, growth factors, and enzymes such as proteases. In fact, the number of mediators is extraordinarily high in MCs as compared with the number of messenger substances known to be formed and released by other cells [16].

On activation, MCs typically release preformed mediators from cytoplasmic granules via regulated exocytosis, but other release options exist [16]. The release of mediator-containing granules, i.e., degranulation, into the extracellular space takes place in a discriminating manner, depending on the stimuli involved and their signaling cascades [16,17,18], inducing multifarious effects on nearby effector cells and the extracellular matrix (ECM) [8,19]. Additionally, direct contact with T-cells and macrophages can promptly release MC mediators [20] and human MCs can constitutively, apparently spontaneously, secrete pro-angiogenic factors [17].

In the mouse, mast cells originate from progenitors made in the extra-embryonic yolk sac starting at embryonic day 8.25 (E8.25) [21,22]. Both extra- and intra-embryonic MC progenitors are generated from endothelial cells [23]. The progenitors are seeded to the fetal tissues and, as differentiated MCs, they are found in almost all vascularized connective tissues, including the central nervous system, in adulthood [13]. So, early MCs are present in fetal tissues before the onset of bone marrow hematopoiesis. The mast cell population is supplemented with mast cell progenitors produced from waves of hematopoiesis that mainly acquire mature characteristics of a mast cell subtype—the mucosal mast cell (MMC) [13]. 

As MCs reside in embryonic tissues from the earliest possible source, it is anticipated that they are required for embryonic and fetal growth since they express significant effectors like growth factors and proteases [13]. In the fetus, MCs may help protect from immunological threats before the adaptive immune is set up and support developmental processes [13]. For instance, fetal MCs express key mediators that control vascular and nerve branching via vascular endothelial growth factor (VEGF) and neurturin [13]; notably, VEGF is a highly specific endothelial cell mitogen and key regulator of angiogenesis. An initial functional link is, thus, created between MCs and endothelial cells since they can communicate reciprocally [7]. This collaboration evidently leads to the formation of blood vessels, the first organ in the embryo, that subsequently develops into the largest network in the body [24]. 

There are diverse and plastic phenotypes of MCs that are influenced by tissue residence. This is because the microenvironment influences their ability to specifically recognize and respond to various stimuli [25]. MCs can undergo alterations in phenotype, anatomical distribution, and numbers following immunological and non-immunological responses. They share a core transcriptional program but differ between many tissues and organs [26,27]. In humans, there are two main types of mast cells, i.e., tryptase- and chymase-positive (MC_TC_) and tryptase-positive (MC_T_). Also, rodents feature two main types of mast cells: the connective tissue mast cell (MC) and the mucosal-type mast cell (MMC) [28]. While MCs contain heparin and relatively high amounts of histamine, MMCs contain over-sulfated glycosaminoglycans and lower amounts of histamine [5,29]. The human MC_TC_ type resembles rodent MCs, while the MC_T_ type corresponds more closely to rodent MMCs. 

### 2.1. Sentinel Cell

Mast cells are located mostly near blood vessels or peripheral nerves and beneath epithelia in connective tissues and mucosal surfaces, allowing them to have a key sentinel role in host defense. Mast cells play important roles in protective immunity against bacteria, viruses, helminth, parasites, fungi, venomous animals, and harmful endogenous and exogenous molecules [7]. The localization of MCs within the blood vessel wall calls attention to their involvement in vasodilatory functions and tissue-specific responses against circulating agents. The presence of MMCs is greatest in the gut, the respiratory system, and the urogenital system, i.e., the internal organs that are most exposed to injurious external factors. Furthermore, MCs control the permeability of blood–brain and blood–cerebrospinal fluid barriers [30]. Mast cells of both main types have a significant role in health and disease as sensors and early responders through interactions with cells in the neighboring tissue, organs, and nervous system [6,7,31]. 

### 2.2. Main Mediators

#### 2.2.1. Biogenic Amines Histamine and Serotonin

Histamine is synthesized and stored in the cytoplasmic granules that contain most of the body’s histamine. The biological impact of histamine depends on its interaction with any of its four G-protein-coupled receptors (H1–H4), which can cause complex physiological and pathological effects like in allergic reactions with profound vasodilatation that can cause cardiovascular collapse. 

Serotonin, produced by many cell types, including MCs of many species, exerts various effects via specific receptors present on numerous types of cells inside and outside the brain.

#### 2.2.2. Proteoglycans

Heparin, a serglycin-carrying glycosaminoglycan heparan sulfate, is exclusively synthesized by MCs [32]. Having the highest negative charge density of any known biological molecule (*Chemical Entities of Biological Interest*: 28304), heparin creates complexes with proteins containing positively charged amino acids, causing conformational and often functional changes in the protein molecule. It plays a central intracellular role, as it is essential in the biogenesis of the cytoplasmic secretory granules that produce and store mediators [33,34,35]. Heparin can also act as an intervenor in cell communication by binding to a variety of signaling molecules [36] and exercises wide-ranging biological functions, including modulation of the release of mediators [32,37,38]. Numerous extracellular proteins, growth factors, cytokines, chemokines, enzymes, and lipoproteins that are involved in a variety of biological processes interact with heparin [39]. Growth factors and cytokines bind to heparin with very high affinity [39]. Moreover, heparin facilitates the interaction between growth factors and their receptors [40,41,42]. 

#### 2.2.3. Proteases

MCs characteristically express tryptases and chymases. Proteases degrade and detoxify venoms, parasites, helminths, harmful microbiological products, microbes, and viruses [43] in addition to their ability to break down proteins into smaller polypeptides or single amino acids and to execute cell signaling. Half of all proteins produced in MCs are proteases [10] that are stored in their fully active form in complex with heparin, ready for rapid release upon MC activation [17]. Most of them are chymotrypsin-related serine proteases, which include tryptases, chymases, cathepsin G, carboxypeptidase A3, etc. [10]. The types, amounts, and properties of the proteases vary depending on mast cell subtype, tissue, and animal species of origin. Proteases modulate growth factors, cytokines, chemokines; generate biologically active fragments from the ECM; ease recruitment of bone-marrow-derived cells; and activate extracellular, highly efficacious, ECM-degrading metalloproteinases (MMPs) [44,45,46]. 

MCs can also produce and release MMPs [47,48], which are one of the most important families of proteinases involved in the tight control of ECM remodeling over time, thereby acting as significant players in connective tissue homeostasis. The ECM plays an important role in regulating angiogenesis as it binds endothelial cells by interacting with integrins present on the cell surface, conducting endothelial cell activity and initiating vascular sprouting.

#### 2.2.4. Cytokines, Chemokines, and Growth Factors

Cytokines are an exceptionally large and diverse group of mostly pro- or anti-inflammatory factors, whereas chemokines are a group of secreted proteins within the cytokine family whose genetic function is to induce cell migration via chemotaxis. Protein growth factors are also included in the cytokine family. MCs store and release multiple types of cytokines, like IL-1 alpha, IL-1 beta, IL-4, IL-6, IL-8, IL-10, IL-17, IL-18, and TNF-alpha, etc. In addition, activated MCs release other de novo synthesized cytokines such as monocyte chemoattractant proteins [9,30,49] and several angiogenic factors like protease 6 and 7, angiopoietins, and MMP-9 [16,17]. Examples of main growth factors expressed by human MCs are basic fibroblast growth factor (bFGF) and several VEGFs that differ in molecular size and in their biological properties. VEGF-A, VEGF-B, VEGF-C, and VEGF-D are expressed by human MCs at both the mRNA and protein level, while the MCs also express mRNA for the VEGF receptors R1 and R2 on the MC surface [50]. MCs are, therefore, a source as well as a target of angiogenic factors, while the release of SCF, VEGF-A 165, VEGF-B 167, VEGF-C, and VEGF-D induce chemotaxis of MCs [50] that causes accretion of MCs at sites of MC secretion, such as MC-mediated angiogenesis. 

#### 2.2.5. Lipid Mediators and Nitric Oxide

Some arachidonic acid metabolites that are newly synthesized by activated MCs execute vascular effects related to angiogenesis, such as the enzymes cyclooxygenase 1 (COX-1, constitutively expressed) and cyclooxygenase 2 (COX-2, inducible), prostaglandins (PGs), thromboxanes, and leukotrienes. COXs are key enzymes in the conversion of arachidonic acid to PGs and other eicosanoids. COX-2 is an important mediator of tumor angiogenesis [51], where its angiogenic effects are mediated primarily by thromboxane A2, PGE2, and PG12. Downstream angiogenesis-related actions of these products include the production of VEGF and induction of MMPs [50]. Moreover, COX-2-derived PGE2 can regulate the angiogenic switch in tumors [52]. Also, COX-2 induces nitric oxide synthase (NOS), the catalyst in the synthesis of nitric oxide (NO); in addition, COX and NOS crosstalk [53]. 

De novo MC-mediated cell proliferation [54] is unaffected by the synthesis of PGs, leukotrienes, or other known arachidonic acid metabolites produced or activated by secreting MCs in the adult rat [55], while the pro-inflammatory PGE2 promotes angiogenesis in the embryonic chick chorioallantoic membrane assay [55]. In adult rats, NO suppresses de novo angiogenesis, mediated by either MC secretion or individual MC-mediators such as IL-1-alpha, TNF-alpha, and bFGF, whilst it lacks effect on de novo VEGF-A-mediated angiogenesis [56,57]. 

### 2.3. Physiological Role Unclear

Despite some 140 years of research that have passed since Paul Ehrlich first described the MC in the late 1870s, its physiological role has not been established [5,8,13,19]. Adaptive MC transfer experiments in genetically modified MC-deficient mouse strains are used to elucidate the physiological role(s) of MCs, although a perfect MC-deficient mouse has not yet been described [5,58]. It has been shown that the MC is essential in mediating allergic reactions and anaphylaxis, while other cells inside and outside the immune system seem to share all the MC’s other functions related to immunological responses [59]. Moreover, MC-deficient mice without defects in the *Kit*-signaling pathway have a remarkably normal immune system [59] and mice virtually lacking MCs develop normally without obvious defects [13]. It is, therefore, unclear if MCs participate in significant developmental processes, while they might be capable of minor adjustments in these processes [13]. So, what enables the preservation of MCs in all vertebrate classes over time? 

### 2.4. Effects on Cell Proliferation, Tissue Remodeling, and Angiogenesis

As first shown in rats, the in situ selectively activated MCs induce potent de novo proliferation in nearby connective tissue and epithelial cells, following non-immunologic or immunologic activation of MCs in different rodent species and tissues [54,60,61].

The body stringently controls angiogenesis by producing an accurate balance of growth and inhibitory factors, also in terms of precise spatial and temporal regulation, in healthy tissues. Angiogenesis, a highly complex process controlled by many pro- and anti-angiogenic factors, is critical for normal physiological development, as in all ages during inflammation, wound healing, tissue repair, ischemia, and pregnancy. The steps required for new vessel development and growth are biologically complex and require coordinated regulation of contributing components, including ECM degradation, modifications of cell-cell interactions, proliferation, migration, and maturation of endothelial and perivascular cells. 

Introducing a biologically sound, highly sensible, discriminating, and quantitative angiogenesis assay exploiting the adult rat mesentery we reported the original finding of de novo MC-mediated angiogenesis [62,63]. This discovery preceded the later knowledge that MCs produce, store, and release many growth factors, such as bFGF, which is a potent angiogenic factor and a powerful inducer of proliferation in a great variety of cells, and VEGF-A, a powerful endothelial cell mitogen and key angiogenic factor.

Histamine and serotonin at very low concentrations are mitogenic, as probably first demonstrated in multilayered, density-inhibited, quiescent human normal fibroblasts in vitro [64]. In rats, released MC histamine in situ exerts a significant de novo mitogenic effect in connective tissue cells via the H2 receptor [65,66] and de novo angiogenesis via the H1 and H2 receptors [67]. A verifying result regarding H1- and H2-receptor-mediated angiogenesis was later reported in mice [68], a study that shows that serotonin is also angiogenic. As is now generally known, histamine is capable of regulating proliferation in different cell populations through any or all of its receptors (H1–H4). Moreover, histamine synergistically promotes bFGF-induced angiogenesis by enhancing VEGF-A production via the H1 receptor [69]. Serotonin exerts mitogenic effects via its specific receptors in many cell types [30,70,71]. 

Heparin initiates proliferation in cultured, multilayered, tissue-like, quiescent human fibroblasts [65,72], which could be explained by the release of heparin-binding growth factor(s) from the ECM and/or facilitation of the interaction between growth factors and their receptors. Heparin, depending on molecule size, affects angiogenesis in a biphasic fashion [12,73,74,75]. The in situ activated rat MCs release very-high-molecular-weight angiogenic heparin. However, due to intrinsic continuous heparinase depolymerization of the heparin, low-molecular-weight anti-angiogenic fragments are created over time, generating an “innate heparin-depolymerization angiogenesis-modulating process” [12]. 

MC-cytokines IL-1-alpha, IL-8, and TNF-alpha induce significant de novo angiogenesis in the rat after being administered at near-physiological dosages [76,77,78]. The inflammatory cytokines conceivably stimulate VEGF-A production in the exposed tissue cells and activate MCs.

#### 2.4.1. Complex Integration of Molecular, Cellular, and ECM Events in MC-Mediated Angiogenesis

Secreting MCs induce and enhance angiogenesis via multiple in-part-interacting avenues including cascade-like paracrine pathways that involve the following: (1) Proteolysis, executed by MMPs and serine proteases, which appears to be one of the first and most sustained activities involved in angiogenesis. A single occasion of MC activation via Compound 48/80 in rats causes electron-microscopic ECM degradation and restructuring, an early sign of tissue remodeling; meanwhile, tissue cells show enhanced metabolism, phagocytosis, collagen synthesis [79], and subsequent cell proliferation [67]. This leads to cellular hypoxia, the induction of transcription hypoxia-inducible factors, and the production of angiogenic factors (e.g., VEGF-A, bFGF, and angiopoietin). This comes in addition to the angiogenic effect of released MC histamine, serotonin, growth factors (e.g., VEGF-A, bFGF, and angiopoietin), and inflammatory cytokines. (2) Release via heparinase, derived from MCs and other activated cells, of growth factors bound to the ECM, which facilitates cell migration and affects the function of cells that interact with the ECM [80]. (3) MC-derived SCF and several VEGFs recruit and activate other cells [7], such as macrophages, platelets, and additional MCs [13,50,81], which can cause accumulation of MCs and macrophages at sites of MC-mediated angiogenesis, and further enhance and prolong the angiogenic process.

#### 2.4.2. Role in Inflammation and Wound Healing

MCs are highly responsive to alarm signals generated after trauma in entering a secretory mode [82]. As is well documented in the literature, MCs are key players in the inflammatory response. Their activity can either promote or suppress inflammatory or immune reactions [5,8,13,83,84,85]. MC-derived TNF-alpha and PGs are, for instance, critical for rapid neutrophil recruitment, whereas other cytokines influence the migration and maturation of antigen-presenting dendritic cells [20]. Enzymatic modifications to ECM moieties, cytokines, and chemokines can induce distinctive cellular responses and are likely part of the mechanism regulating the perpetuation or arrest of inflammation [9,11,30,45]. 

The activated MC plays a key role in wound healing as it controls all critical phases, i.e., inflammation, proliferation, and remodeling. This is performed by triggering and modulating the inflammatory stage, inducing proliferation and collagen production of connective tissue cells, inducing angiogenesis, and executing remodeling of the ECM [86,87,88,89,90,91,92,93]. Findings in the rat mesentery of MC-mediated [54] and MC-histamine-mediated [66] cell proliferation and, likewise, angiogenesis [63,67] are in keeping with the subsequent discoveries in a mouse model for MC deficiency that both MC-activation and histamine release are required for normal cutaneous wound healing [93]. In later phases of skin wound healing, histamine and serotonin stimulate epidermal keratinocyte proliferation, while TNF-alpha has an inhibitory effect [94].

### 2.5. Role in Reproduction

MCs are hormonally affected by the female sex system and are activated during different phases of the menstrual cycle, including an extensive degranulation before and during menstruation [10,95]. MCs are, moreover, essential modulators of the immune response during pregnancy and for successful pregnancy. This topic is extensively reported [9,10,95,96]. 

MCs are found in the ovaries, myometrium, and endometrium in humans and rodents. The number of ovarian MCs is regulated positively by local gonadotrophin-releasing hormone expression and negatively by prolactin [97]. Estrogen induces MC degranulation in the ovaries. In ovulation, angiogenesis is a crucial step [98] involving angiogenic factors like bFGF, VEGF-A, and histamine. Ovulation enables fertilization in the oviduct, as shown in rodents [10,96,99,100,101]. Uterine MCs express estrogen and progesterone surface receptors. These two main female sex hormones influence the migration of MCs to the uterus and cause their maturation and activation in the mouse uterus [102]. Uterine MCs are a heterogenous population consisting of MCs, MMCs, and cells that present characteristics of both types.

In the pregnant uterus, MC activation creates changes in the fetal maternal interface, tissue remodeling, spiral artery modifications, angiogenesis, and alterations in the muscular wall [89,96,101,102,103]. MC secretion is required for the development of the decidua, as for vascularization, cell proliferation, and growth of the placenta [96]. In addition, MCs seem to control uterus contractions that are important for giving birth [9], while retaining their protection against pathogens [9]. Postnatally, MC proteases may be involved in uterine tissue remodeling [104]. 

In human testis, but not in rat and mouse testis, MCs usually appear in the interstitial compartment [105]. The roles of MCs in the male reproductive system, including spermatogenesis, are less well understood, although both MC/MC_TC_ and MMC/MC_T_ are present in the testis and epididymis [89,106]. 

### 2.6. Role in Health and Homeostasis

MCs react faster than other tissue-resident immune cells to induce an initial immune response following invasion of pathogens or other harmful factors through a wide array of receptors like alarmins, purinergic, and pattern recognition receptors [7,9,107]. The ability of the MC to detect changes in the microenvironment and respond while maintaining adjustable mobility and adaptability enables this “master cell” [8] to play a leading role in the complex processes of immune homeostasis and normal tissue functioning. MCs execute this by engaging in crosstalk with peripheral nerves and causing cell activation, chemotaxis, ECM degradation and restructuring, cell proliferation, vascular leakage, and angiogenesis [7,12,13]. The ways in which the MC exerts profound functions in immune regulation, tissue homeostasis, and health have recently been reviewed broadly [7,9,10,11,13,20]. 

Tissue homeostasis is, obviously, any self-regulating process by which an organism tends to maintain microenvironmental, tissue, and organ stability while adjusting to conditions that are best for its health and survival. The MCs are recognized as important housekeeping cells causing homeostatic effects in the microenvironment of tissues and organs [27]. As observed in many tissues and organs [108,109,110], there is an age-related increase in MC number and activity.
ROLES OF CONNECTIVE TISSUE MAST CELLS IN SUCCEEDING GENERATIONSCONTRIBUTING TO PERPETUATION OF MAMMALS AND OTHER VERTEBRATES  GENERATION #n1.1 Embryonic and Fetal LifeMCs are generated from endothelial cells *** and participate in the creation of the first organ **, blood vessels, that later develops into the body’s largest network. Critical role in inflammation; potentially lethal **. Critical role in wound healing/tissue repair **.1.2 Birth throughout LifeCritical role in commonplace inflammation; potentially lethal *******. Critical role in commonplace wound healing/tissue repair; potentially lethal *******.1.3 Female Sex System beginning at MenarcheOvulation *******. Pregnancy *******. Childbirth *****.GENERATION #n + 12.1 Embryonic and Fetal LifeMCs are generated from endothelial cells ******* and participate in the creation of the first organ ******, blood vessels, that later develops into the body’s largest network. For any newly created embryo, go to 1.1 above and there is an endless loop: 2.1, 2.2, 2.3, 3.1, 3.2, etc. Table. Connective tissue mast cells (MCs) are definitely *******, probably ******, or likely ***** key players in the creation of a permanent loop of life-promoting and life-sustaining events in succeeding generations, safe-guarding the offspring. References are given in the text.

## 3. Discussion and Conclusions

In the embryo, mast cell progenitors are generated from endothelial cells [22]. These early MCs express the angiogenic cytokine VEGF, heparin, and proteases and communicate reciprocally with endothelial cells [4,7,13]. This evidently leads to the creation of the first organ in the embryo, the blood vessels, which later develops into the largest network in the body [24]. The MC, a preeminent sentinel cell, plays important roles in protective immunity, health, and tissue homeostasis. The MC is, furthermore, decisive in fundamental life-promoting and life-preserving events like ovulation, pregnancy, inflammation, and wound healing. A basic process in these MC-dependent outcomes seems to be the serial effects that follow upon MC secretion in situ, like the activation of non-MCs, ECM and tissue remodeling, tissue–cell proliferation, MC and non-MC chemotaxis, and angiogenesis [12]. MC-mediated angiogenesis [63] is a constant feature in these life-maintaining events. Our understanding of the data is that the MC enables the perpetuation of mammals and vertebrates in general. This is because of its probably critical role in the creation of the first organ, i.e., the blood vessels, as well as its key roles in reproduction, inflammation, and wound healing from birth up to reproductive age. Hence, an unending life-sustaining loop is created, safeguarding the offspring.

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
