# Peer review of "On Connective Tissue Mast Cells as Protectors of Life, Reproduction, and Progeny"

_ijms, 2024, doi:10.3390/ijms25084499_

Round 1
Reviewer 1 Report
Comments and Suggestions for Authors
1. The title of the manuscript may be confusing to readers because the manuscript generally focuses on connective tissue mast cells. Therefore, the title of the manuscript must be changed.
2. Keywords usually do not include words that appear in the title. In what order are keywords written?
3. The use of abbreviation MC is confusing. MCs in lines 27-43 are connective tissue mast cells, MCs in lines 45-84 are mast cells, and MCs in lines 85-95 are mast cells and connective tissue mast cells. Right, did I understand it correctly? Are the MCs in line 97, 101, 105 and 106 mast cells or connective tissue mast cells ?
4. What do inside and outside mean lines 192-194 ?
5. The contents of 2.4., 2.4.1, and 2.5 include “Effects on cell proliferation, tissue remodeling and angiogenesis”. So you need to change 2.5 to 2.4.2.
6. The format of Table 1 and Table 2 needs to be changed. This type of table is generally not helpful to the reader. The table must be changed to a form that is intuitively understandable.
Author Response
Reviewer 1
The point made that the manuscript does not significantly improve our knowledge about mast cells, could, with all due respect, be applied broadly to review articles as original findings are usually published elsewhere. Evidently, well-documented reviews are helpful particularly to readers that are not super-specialists of the field discussed. As clearly stated in the text, the subject expands on a recent novel concept (APMIS, 130, 618-624, 2022). The argument presented in the present manuscript of the connective-tissue mast cell having a significant role in mammalian perpetuity, including a suggested mechanism for safeguarding the offspring, is certainly of interest from a biological point of view. This has apparently not previously been put forward so distinctly as done here, together with an update of the mast cell’s well-known other life-promoting influences. The fact that embryonic mast cells are generated from endothelial cells, and that these cells by communicating reciprocally probably generate the blood vascular system, fits well in. It should strengthen the concept that the mast cell is apparently basically a survival cell because of its key actions at various strategical locations in significant physiological and pathological processes.

Reviewer 2 Report
Comments and Suggestions for Authors
The Author of this review article has analyzed the role of mast cells in different physiological and pathological conditions, involving tissue remodeling, cell proliferation, angiogenesis, inflammation, wound healing, tissue homeostasis, and reproduction. Remarks. Even if this MS is well-written and documented, it doesn't significantly improve our knowledge about mast cells and overlaps with another review article recently published by the same Author (APMIS, 130: 618-624, 2022).
Author Response
Reviewer 2
- The point made about the title is well taken. The title is changed accordantly:
ON CONNECTIVE-TISSUE MAST CELLS AS PROTECTORS OF LIFE, REPRODUCTION, AND PROGENY
- Keywords are set in alphabetic order and do not contain words that appear in the title.
- Your point is well taken. Corrections are made accordingly: (a) in lines 27-43, MCs are connective tissue mast cells, (b) in line 45, The MC .. is changed to The mast cell, (c) line 69, So, MCs are present .. is changed into So, early MCs are present, (d) line 70, The MC population is further supplemented .. is changed to The MC population is supplemented .., (e) line 97/98, MCs .. are changed to Mast cells.., (f) line 100/101, MCs are MCs, (g) line 106/107, MCs .. is changed to Mast cells.
- The phrase ‘inside and outside’ is a quote from the reference given. This has been deleted in the Discussion and Conclusion section.
- The contents of 2.4, 2.4.1, and 2.5 is changed according to your suggestion, e., to delete 2.5 in exchange for 2.4.2. Therefore, 2.6 is changed to 2.5, etc.
- There was no table in my initially submitted manuscript. However, the then-assigned Editor demanded a table and/or figure. This was, I suppose, because the Editor felt that a table would help the reader to follow the reasoning. I agree that Table 1, which was meant as an introduction to Table 2, summing up data that are clearly presented in the text, can be deleted. Table 2 (Table), standing alone, is redesigned and comes with much-needed explanatory caption. I believe that it is easily read while clarifying, with varying estimated degrees of certainty, the precise claimed generation-bridging effects of the connective-tissue mast cells.
Round 2
Reviewer 1 Report
Comments and Suggestions for Authors
In mammals, mast cells of connective tissue are generated from embryonic endothelial cells and not only are responsible for immunity, but also play an essential role in maintaining tissue homeostasis. Their role from birth to reproduction is well explained to the journal readers.
Reviewer 2 Report
Comments and Suggestions for Authors
The Author has improved the ms.